# Predicting Age-related Macular Degeneration Progression from Retinal Optical Coherence Tomography with Intra-Subject Temporal Consistency

**Arunava Chakravarty**[1]           ARUNAVA.CHAKRAVARTY@MEDUNIWIEN.AC.AT
**Taha Emre**[1]           TAHA.EMRE@MEDUNIWIEN.AC.AT
**Dmitrii Lachinov**[1,2]           DMITRII.LACHINOV@MEDUNIWIEN.AC.AT
**Antoine Rivail**[1,2]           ANTOINE.RIVAIL@MEDUNIWIEN.AC.AT
**Ursula Schmidt-Erfurth**[1]           URSULA.SCHMIDT-ERFURTH@MEDUNIWIEN.AC.AT
**Hrvoje Bogunović**[1,2]           HRVOJE.BOGUNOVIC@MEDUNIWIEN.AC.AT

[1] *OPTIMA Lab, Department of Ophthalmology, Medical University of Vienna, Austria*

[2] *Christian Doppler Lab for Artificial Intelligence in Retina, Medical University of Vienna, Austria*

**Editors:** Accepted for publication at MIDL 2024

## Abstract

The wide variability in the progression rates of Age-Related Macular Degeneration (AMD) and the absence of well-established clinical biomarkers make it difficult to predict an individual's risk of AMD progression from intermediate stage (iAMD) to late dry stage (dAMD) using Optical Coherence Tomography (OCT) scans. To address this challenge, we propose to jointly train an AMD stage classifier to discriminate between iAMD and dAMD with a N-ODE that models the future trajectory of the disease progression in the learned embedding space. A temporal ordering is imposed such that the distance of a scan from the decision hyperplane of the AMD stage classifier is inversely related to its time-to-conversion. In addition, an intra-subject temporal consistency in the predicted conversion risk scores is ensured by incorporating a pair of longitudinal scans from the same eye during training. We evaluated our proposed method on a longitudinal dataset comprising 235 eyes (3,534 OCT scans) with 40 converters. The results demonstrate the effectiveness of our approach, achieving an average area under the ROC of 0.84 for predicting conversion within the next 6, 12, 18 and 24 months. Additionally, the Concordance Index of 0.78 surpasses the performance of several popular methods for survival analysis.

**Keywords:** Survival Analysis, AMD, OCT, Longitudinal disease progression, Retina

## 1. Introduction

Age-related macular degeneration (AMD) is a leading cause of blindness among the elderly population (Wong et al., 2014). It is asymptomatic in its early and intermediate stages (iAMD), characterized by the presence of drusen. AMD gradually advances to the late stage leading to irreversible vision loss which could be categorized as either neovascular (nAMD) or dry (dAMD). nAMD is caused by abnormal blood vessel growth in the choroid that leaks fluid into the retina. dAMD is more prevalent than nAMD and characterized by Geographic Atrophy (GA) due to the loss of Retinal Pigment Epithelium (RPE). Recently, for the first time, drugs for dAMD (Khanani et al., 2023; Heier et al., 2023) were approved by FDA. Patients in the iAMD stage are regularly monitored with longitudinal Optical

Coherence Tomography (OCT) imaging across multiple visits to initiate treatment at the earliest onset of late AMD to minimize vision loss. Identifying iAMD patients at a high risk of dAMD conversion enables ophthalmologists to prioritize these cases for enhanced monitoring, facilitating early detection of dAMD onset. However, this is a challenging task due to the absence of well-established clinical biomarkers and significant inter-subject variations in the rate of AMD progression. Deep learning (DL) methods to predict the future risk of conversion of an eye from iAMD to dAMD can play a critical clinical role in supporting personalized treatments and clinical research by categorizing iAMD patients into distinct risk levels for biomarker identification and recruitment in clinical trials.

**Related Work:** Existing methods for predicting the risk of conversion from iAMD to nAMD or dAMD fall into two main categories: biomarker and image-based approaches. *Biomarker-based* methods (Sleiman et al., 2017; Schmidt-Erfurth et al., 2018; Banerjee et al., 2020; de Sisternes et al., 2014; Lad et al., 2022) involve segmenting retinal tissues and pathologies to extract features, subsequently combined with clinical and demographic data for risk prediction. Notably, Banerjee et al. (2020) utilizes multiple past visit biomarkers in an LSTM network for future risk assessment. *Image-based* methods, however, directly utilize DL models on raw OCT scans, bypassing manual segmentation. A hybrid approach using both biomarker and image features for predicting nAMD conversion is presented in Yim et al. (2020), employing an ensemble DL model. Unlabeled longitudinal OCT datasets have been used in Emre et al. (2022); Rivail et al. (2019) for feature learning via temporal self-supervised learning. These methods typically employ a binary classifier for predicting conversion within specific timeframes, (e.g., 2 years (Russakoff et al., 2019), 6 months (Yim et al., 2020; Emre et al., 2022)), or multi-label classification for various discrete time-intervals (e.g., 6, 12, and 18 months (Rivail et al., 2019)).

The binary classification based approaches are limited by discretization of the conversion time and their inability to manage censoring, which occurs when an eye's actual conversion time is unknown due to missing follow-ups or non-conversion within a limited study duration. Survival analysis addresses these challenges. Discrete survival models are similar to multi-label classification but modify training loss to incorporate censoring and have recently been applied to predict dAMD conversion (Rivail et al., 2023). A transformer model has also been used for discrete-time modeling of the hazard function from tabular clinical and demographic data (Hu et al., 2021). Traditional non-DL continuous models of survival analysis have also been explored to capture AMD progression with handcrafted biomarkers using the linear Cox Proportional Hazard model (CoxPH) (Schmidt-Erfurth et al., 2018). Although CoxPH has been extended with DL using images (Katzman et al., 2018), they have not yet been explored to model AMD progression so far. Moreover, these models are inflexible as each patient's hazard function is constrained to be a scaled version of the same baseline hazard across the entire population. SODEN (Tang et al., 2022) overcomes this issue by employing a N-ODE to model the cumulative hazard function for survival on tabular data. The GRU-ODE-Bayes (De Brouwer et al., 2019) proposed a N-ODE to extend the GRU based Recurrent Neural Network in continuous time, used in predicting disability progression in Multiple Sclerosis patients from tabular data of past history. Recently, N-ODEs have also been used to model the spatial evolution of GA segmentation in OCT (Lachinov et al., 2023) and Diabetic retinopathy in fundus images (Zeghlache et al., 2023).

**Contributions:** Our key contributions are: (i) The time-to-conversion from iAMD to nAMD is modeled in continuous time, rather than discrete time-intervals as used in most existing methods. Our model can therefore use actual continuous conversion times as ground-truths during training and also predict conversion probabilities within arbitrary continuous times. (ii) Our novel N-ODE based modeling directly models the Cumulative Distribution Function(CDF) of the future conversion time instead of the cumulative hazard function used in existing methods like SODEN. Our SMGRU-ODE architecture also extends ODE-GRU by stacking multiple layers with multiple parallel heads. (iii) We incorporate intra-subject consistency by requiring the N-ODE estimates of the feature and risk at future time-points to be consistent with the values obtained using the actual OCT scan of the future visit. (iv) We jointly train a linear AMD stage classifier and employ a rank loss on its logits which is sensitive to censoring, to regularize the feature embedding. This facilitates patient stratification into risk groups based on a scalar risk score derived from the decision hyperplane distance for clinical studies or personalized treatment.

## 2. Method

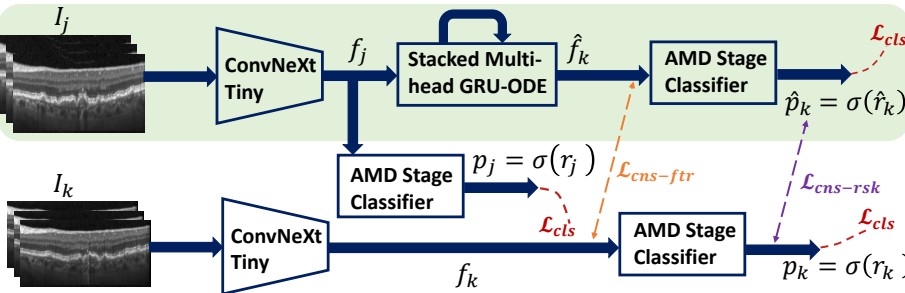

Figure 1: Siamese architecture uses shared weights for the ConvNeXt-Tiny encoder and linear AMD stage classifier in both branches. Top branch predicts current AMD stage at time-point $t = j$ and evolves features with GRU-ODE for future $t = k$. Bottom branch computes features and stage predictions for $t = k$ directly from the future scan $\boldsymbol{I}_k$. Losses $\mathcal{L}_{cns-ftr}$ and $\mathcal{L}_{cns-rnk}$ ensure consistency in the feature and risk predictions between branches. $\mathcal{L}_{rnk}$ loss ranks the logit $r$ in inverse order of conversion time. Only the top branch (shaded in green) is used for inference.

Given an input OCT scan of an iAMD patient, the proposed method projects it to a feature embedding where the current feature is evolved over time with a N-ODE to forecast the future trajectory of the disease progression. The features estimated for any (continuous) future time-point are fed to a linear AMD stage classifier to predict the probability of the eye to have already converted within that time, thereby modeling the CDF of the future conversion time. In Survival analysis, the Ground Truth (GT) label for an OCT image $\boldsymbol{I}_j$ is defined by the tuple $(E_j, T_j)$. The event indicator $E_j = 1$ signifies that the eye associated with scan $\boldsymbol{I}_j$ will progress from iAMD to dAMD, while $E_j = 0$ denotes no conversion within the monitoring period. $T_j$ denotes the time of conversion from the current visit if $E_j = 1$

or the censoring time until when the patient was last monitored. Our approach is trained on batches comprising random image pairs $\boldsymbol{I}_j$, $\boldsymbol{I}_k$ (Figure 1) of the same eye captured at different time points $t_j, t_k \in \mathbb{R}_{\geq 0}$ from two visits, such that $\boldsymbol{I}_j$ precedes $\boldsymbol{I}_k$ with $t_j < t_k$.

**AMD Stage Classifier:** Both $\boldsymbol{I}_j$, $\boldsymbol{I}_k$ are input to the same ConvNeXt-Tiny Encoder to obtain the features $\boldsymbol{f}_j$ and $\boldsymbol{f}_k$ respectively. ConvNext-Tiny with 29M parameters and 4.5GFLOPs is comparable to ResNet-50 and outperforms similar-sized Vision Transformer (ViT) architectures (Liu et al., 2022), making it a suitable choice for our task. The stage classifier's GT $y_j^{cls} = 1$ if both $T_j \leq 0$ and $E_j = 1$, otherwise $y_j^{cls} = 0$ for the scan $\boldsymbol{I}_j$. The stage classifier predicts the logit $r_j$. The probability for the current AMD stage for $\boldsymbol{I}_j$ is $\boldsymbol{p}_j = \sigma(r_j)$, where $\sigma(.)$ denotes the sigmoid activation. Notably, $r_j$ is proportional to the distance of $\boldsymbol{f}_j$ from the decision hyperplane of the AMD stage classifier and would be used below to define a risk score for future conversion. The stage classifier treats each scan independently without considering any correlations between two scans of the same eye from different time-points. While it enables the learned feature to capture pathologies to distinguish dAMD from iAMD, it may fail to capture more subtle retinal changes indicative of how AMD will progress in the future (Appendix Figure 3(a)).

**Time-Series Prediction:** To address these issues, we incorporate a N-ODE based continuous time-series predictor called Stacked Multihead GRU-ODE (SMGRU-ODE) to model the future trajectory of AMD progression in the feature embedding using the current scan. SMGRU-ODE evolves the current feature $\boldsymbol{f}_j$ over a $(t_k - t_j)$ time-interval to independently predict the future feature $\hat{\boldsymbol{f}}_k$ for time $t_k$ directly from the prior visit $\boldsymbol{I}_j$, while the actual feature $\boldsymbol{f}_k$ is also obtained from $\boldsymbol{I}_k$. The encoder, SMGRU-ODE and stage classifier can now be jointly trained with the AMD stage classification task:

$$\mathcal{L}_{cls} = L_{bce}\left(y_j^{cls}, p_j\right) + L_{bce}\left(y_k^{cls}, p_k\right) + L_{bce}\left(y_k^{cls}, \hat{p}_k\right), \tag{1}$$

where $L_{bce}(y, p)$ is the binary cross-entropy loss. $p_j$, $p_k$ and $\hat{p}_k$ are predictions from the stage classifier for the features $\boldsymbol{f}_j$, $\boldsymbol{f}_k$ and $\hat{\boldsymbol{f}}_k$ respectively. The SMGRU-ODE architecture is detailed in Section 2.1

**Intra-eye Consistency:** For the disease progression trajectory predicted by the SMGRU-ODE to be consistent, the features $\hat{\boldsymbol{f}}_k$ and its stage prediction $\hat{p}_k$ should match the corresponding $\boldsymbol{f}_k$ and $p_k$, obtained directly from $\boldsymbol{I}_k$. This consistency loss between the features $(\mathcal{L}_{cns-ftr})$ and the stage predictions $(\mathcal{L}_{cns-rsk})$ are defined as:

$$\mathcal{L}_{cns-ftr} = ||\boldsymbol{f}_k - \hat{\boldsymbol{f}}_k||_2^2, \qquad\qquad \mathcal{L}_{cns-rsk} = L_{bce}(p_k, \hat{p}_k). \tag{2}$$

These losses combined with $\mathcal{L}_{cls}$ ensure that the learned feature embedding (Appendix Figure 3 (a) vs (b)) not only characterizes the current disease stage but is also sensitive to the subtle retinal changes that capture the trajectory of the disease progression in the future. Since the future visit scans are unavailable at test time, only the top branch in Figure 1 highlighted in green is employed to obtain future predictions with the N-ODE.

**Risk Score Ranking:** $\mathcal{L}_{cls}$ ensures that the iAMD and nAMD samples lie on opposite sides of its decision hyperplane without imposing any ordering among iAMD cases. We envision a regularized feature manifold (Appendix Figure 3(b) vs (c)) which correlates the risk of disease progression of a feature point to be inversely related to its distance from decision hyperplane, i.e., the closer an iAMD sample is to the decision hyperplane, the

smaller its time to conversion, until it crosses over the hyperplane to the dAMD class. In this case, the logits $r$ from the stage classifier acts as a risk score for AMD progression as it is proportional to the distance of the sample from the decision hyperplane. While predicting the probability of conversion within specified time-points (CDF) requires the N-ODE during inference, a scalar risk score is directly obtained from the logits of the stage classifier from the current scan. We consider a loss $\mathcal{L}_{rank}$ which is defined using pairs of samples to encourage such ordering. Given a training batch comprising $B$ pairs of images (i.e., a total of $2 \times B$ scans), we form all possible pairs $(\boldsymbol{I}_m, \boldsymbol{I}_n)$ which may or may not come from the same eye. $\mathcal{L}_{rank}$ defines an auxiliary classification task where the difference of their scalar logits from the AMD stage classifier is fed through a neuron (with a single input and output) to obtain the probability of ranking $r_m > r_n$ as $P_{m>n} = \sigma \left( w \cdot (r_m - r_n) + b \right)$, where $w$ and $b$ are scalar parameters of the neuron and the loss is defined as $\mathcal{L}_{rank} = L_{bce} \left( y_{rank}^{m>n}, P_{m>n} \right)$.

The GT $y_{rank}^{m>n} = 1$, if $T_m < T_n$ and $E_m = 1$ (indicating the $\boldsymbol{I}_m$ converts before $\boldsymbol{I}_n$) or when $T_m < T_n$ and both $\boldsymbol{I}_m, \boldsymbol{I}_n$ are scans of the same eye (the risk increases in the future visits as damage to the retinal tissue is irreversible). Similarly, $y_{rank}^{m>n} = 0$ if $T_m > T_n$ and, either $E_n = 1$ or $\boldsymbol{I}_m, \boldsymbol{I}_n$ come from the same eye, which signify cases where $\boldsymbol{I}_n$ converts before $\boldsymbol{I}_m$. The image pairs that do not fall into either one of these two categories cannot be ranked due to censoring and are considered to have missing labels that are masked out during the loss computation. Finally, the total loss to train the proposed model is:

$$\mathcal{L}_{tot} = \lambda_1 \mathcal{L}_{cls} + \lambda_2 \mathcal{L}_{cns-rsk} + \lambda_3 \mathcal{L}_{cns-ftr} + \lambda_4 \mathcal{L}_{rank}, \tag{3}$$

where the loss weights $\lambda_1$, $\lambda_2$, $\lambda_3$ and $\lambda_4$ are not handcrafted but dynamically adapted during training using MTAdam (Malkiel and Wolf, 2021) (see Appendix E for more details).

## 2.1. The N-ODE architecture

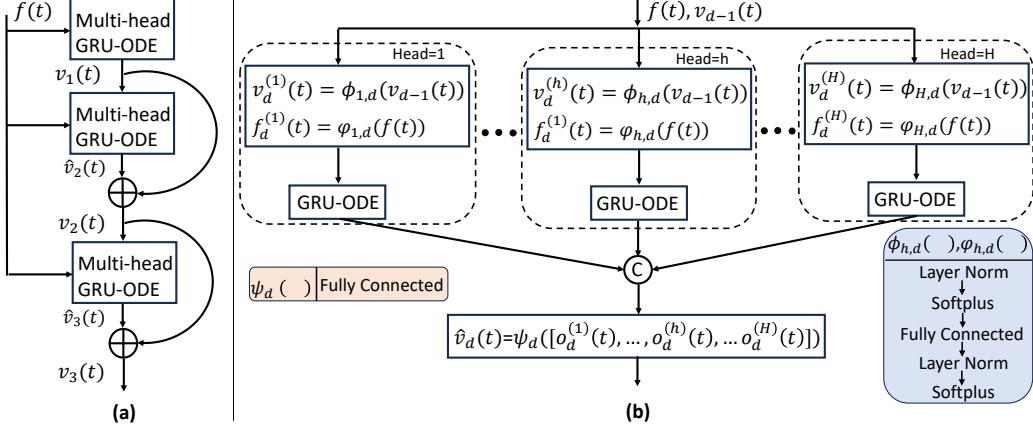

Figure 2: Our SMGRU-ODE extends GRU-ODE (in eq. 4) by stacking multiple layers (a), with multiple parallel heads in each layer (b).

Upon projecting the initial scan $\boldsymbol{I}_j$ to the feature $\boldsymbol{f}_j$, the N-ODE predicts its future trajectory in the feature embedding to model disease progression. Let $\boldsymbol{f}(t)$ represent the

feature after a time $t$ has elapsed since $\boldsymbol{I}_j$ was imaged. As time progresses from $t$ to $t + dt$ by an infinitesimal amount, $\boldsymbol{f}(t)$ is displaced by $\boldsymbol{v}_D \cdot dt$ where $\boldsymbol{v}_D$ denotes the instantaneous velocity vector. This can be modeled in continuous time using the N-ODE $\frac{d\boldsymbol{f}(t)}{dt} = \boldsymbol{v}_D(\boldsymbol{f}(t))$ with the initial value $\boldsymbol{f}(0) = \boldsymbol{f}_j$, where $\boldsymbol{v}_D$ is modeled with a DL network. We assume a time-invariant system, i.e., $\boldsymbol{v}_D$ is solely dependent on the current feature $f(t)$ and not on the time $t$ elapsed so far.

We propose the SMGRU-ODE network to model $\boldsymbol{v}_D(\boldsymbol{f})$ which extends GRU-ODE (De Brouwer et al., 2019) by stacking $D = 3$ layers (Figure 2(a)), and modifying each layer to have $H = 12$ parallel pathways (Figure 2(b)) called heads, based on the efficacy of such design in Vision Transformers (Dosovitskiy et al., 2020) and recent CNN architectures (Liu et al., 2022), (Xie et al., 2017). In Figure 2(a), each layer employs $\boldsymbol{f}(t)$ as the hidden state and except for the first layer, also accepts an external input $\boldsymbol{v}_{d-1}(t)$ from its previous layer. Additive skip residual connections are applied between the inputs and outputs of each layer $(\boldsymbol{v}_d(t) = \hat{\boldsymbol{v}}_d(t) + \boldsymbol{v}_{d-1}(t))$. As depicted in Figure 2(b), each head $1 \le h \le H$ independently projects the two, $d$-dimensional (768 for ConvNeXT-Tiny (Liu et al., 2022)) inputs to a $d/H$ dimensional sub-space using the fully connected (FC) layers, $\varphi_{h,d}(\boldsymbol{f}(t))$ and $\phi_{h,d}(\boldsymbol{v}_{d-1}(t))$ which project $\boldsymbol{f}(t)$ and $\boldsymbol{v}_{d-1}(t)$ to $\boldsymbol{f}_d^{(h)}(t)$ and $\boldsymbol{v}_d^{(h)}(t)$ respectively. Next, the $h^{th}$ head computes the output $\boldsymbol{o}_d^{(h)}(t)$ similar to GRU-ODE as

$$\boldsymbol{r}_d^{(h)}(t) = \boldsymbol{\Psi}_{h,d}^{(rst)}\left(\left[\boldsymbol{v}_d^{(h)}(t), \boldsymbol{f}_d^{(h)}(t)\right]\right), \quad \boldsymbol{u}_d^{(h)}(t) = \boldsymbol{\Psi}_{h,d}^{(updt)}\left(\left[\boldsymbol{v}_d^{(h)}(t), \boldsymbol{f}_d^{(h)}(t)\right]\right), \tag{4a}$$

$$\boldsymbol{g}_d^{(h)}(t) = \boldsymbol{\Psi}_{h,d}^{(act)}\left(\left(\boldsymbol{r}_d^{(h)}(t) \odot \boldsymbol{f}_d^{(h)}(t)\right)\right), \quad \boldsymbol{o}_d^{(h)}(t) = \left(1 - \boldsymbol{u}_d^{(h)}(t)\right) \odot \left(\boldsymbol{g}_d^{(h)}(t) - \boldsymbol{f}_d^{(h)}(t)\right) \tag{4b}$$

where $\boldsymbol{\Psi}_{h,d}^{(rst)}(.)$ and $\boldsymbol{\Psi}_{h,d}^{(updt)}(.)$ comprise a FC layer followed by Layer Normalization (LN) and sigmoid activation to compute the *update* and *reset gates*, $\boldsymbol{u}_d^{(h)}(t)$ and $\boldsymbol{r}_d^{(h)}(t)$ in eq. (4) respectively. The $\boldsymbol{\Psi}_{h,d}^{(act)}(.)$ used for the *candidate activation vector* $\boldsymbol{g}_d^{(h)}(t)$ employs a Softplus activation after the LN and FC layers. Finally, each head's output $\boldsymbol{o}_d^{(h)}(t)$ is concatenated and input to $\Psi_d(.)$, which represents a FC layer *without* LN and activation, to obtain $\hat{\boldsymbol{v}}_d(t)$. The LN and activations are instead applied at the beginning of the next layer in $\varphi_{h,d+1}(.)$ and $\phi_{h,d+1}(.)$. This is to ensure that (i) the output of the final layer $\boldsymbol{v}_D$ can take arbitrary (including negative) values and (ii) the normalization and activation is applied after the additive residual connections in each layer (Figure 2(a)) so that backpropagation gradients can be improved through pre-activation (He et al., 2016). During forward pass of the N-ODE, the feature for a future time-point $k$ is given by $\hat{\boldsymbol{f}}_k = \boldsymbol{f}_j + \int_0^{k-j} \boldsymbol{v}_D(f_{t+j})dt$ which can be numerically estimated using any black-box ODE-solver. During the backward pass, the computational graph related to each iteration of the ODE-solver is not saved but can be estimated on the run by solving another augmented ODE introduced by the adjoint sensitivity analysis in (Chen et al., 2018). As a result, the training requires a constant amount of memory independent of the solver's step size and integration time allowing us to evolve the trajectory over long time-intervals, even with limited GPU memory.

## 3. Experiments and Results

**Dataset:** It consists of 3,534 OCT scans from 235 eyes (40 converters and 195 censored) from 123 patients, collected at the Department of Ophthalmology, Medical University of

Vienna (Schlanitz et al., 2017) and acquired using a Spectralis scanner at a resolution of 49 B-scans (slices), each with a $512 - 1024 \times 496$ pixels. Each eye was imaged every 3-6 months, with total follow-up periods spanning 2-7 years. For converter eyes, labels for each scan were computed by measuring the time interval between its acquisition and the first conversion visit. Our Pytorch code is available at `https://github.com/arunava555/Multihead_GRU_ODE_based_Survival_Analysis`.

**Experimental Setup:** A stratified five-fold cross-validation was performed by randomly dividing the scans at an eye-level to reduce the bias of a specific train-test data split. Each fold had 47 eyes with 8 converters and the number of scans varied between 667-707 across the folds. The model was trained five times, treating each fold as the test set, while the remaining dataset was randomly divided into 80% for training and 20% for validation. While the already converted dAMD scans were used during training, they were removed from the test set during evaluation. The performance was evaluated for predicting the conversion to dAMD within 6, 12, 18 and 24 months using the Area under the receiver operating characteristic curve (AUROC). Balanced Accuracy was used to assess binary predictions obtained by setting a threshold on the conversion probabilities at an optimal operating point determined from the validation set in each fold. Additionally, the Concordance Index (C-index) was used to evaluate the proposed risk score. It quantifies a model's ability to provide a reliable (inverse) ranking of the conversion time, taking censoring into account.

**Ablation Results:** In Table 1, we analyzed the effect of the depth and the number of heads in SMGRU-ODE. Either reducing the depth D from 3 to 1 (in row 1) while keeping H=12, or reducing H from 12 to 1, while keeping D=3 (in row 2) had an adverse impact on the performance at all time-points, both in terms of AUROC and Balanced Accuracy. The C-index also reduced from 0.777 to 0.744 in both cases. This justifies our incorporation of multiple layers and heads in GRU-ODE. From rows 3-6, we perform ablation on the loss terms. In row 3, we train the model with $\mathcal{L}_{cls}$ loss (see eq. (1) ), which is the minimal loss required to predict future conversion without incorporating any other losses to regularize the feature embedding. Introducing $\mathcal{L}_{rank}$ loss to it leads to a significant improvement in C-index (from 0.715 to 0.769) which is expected as $\mathcal{L}_{rank}$ is geared towards improving the rank ordering. Moreover, it improves the conversion prediction performance for all time-points both in terms of AUROC and Balanced Accuracy (except for AUROC - 6 month). Next, introducing the $\mathcal{L}_{cns-rsk}$ loss (in row 5) leads to further improvement in C-index, AUROC also improves for all except the 24-month time-point. However, the Balanced Accuracy shows mixed results with minor improvements for predicting conversion within 12 and 18 months but a slight drop in performance for the 6 and 24-month time-points. Finally, introducing the $\mathcal{L}_{cns-ftr}$ loss leads to our proposed method in row 6. It consistently improves the AUROC, Balanced Accuracy and C-index metrics with the exception of the 6 month time-point. Overall, the results demonstrate the value of using all loss terms.

**Comparison with the State of the Art:** In Table 2, we compare our method against common survival analysis methods. 6-month time windows are considered for the discrete-time survival models based on the censored cross-entropy loss (Wulczyn et al., 2020) and the logistic hazard model (Rivail et al., 2023). DeepSurv (Katzman et al., 2018) extends CoxPH with DL, while SODEN (Tang et al., 2022) is a N-ODE based method, previously used on tabular data. These methods were also trained with ConvNeXt-Tiny encoder but with modified classification layers and losses. Notably, all of these methods do not employ intra-

Table 1: Ablation experiments of different loss terms and the SMGRU-ODE architecture (mean ± std. dev.). Best values in each column are highlighted in bold.

| | AUROC | | | | Balanced Accuracy | | | | |
| | 6 | 12 | 18 | 24 | 6 | 12 | 18 | 24 | C-index |
|---|---|---|---|---|---|---|---|---|---|
| SMGRU-ODE(D=1) | $0.823 \pm 0.05$ | $0.789 \pm 0.05$ | $0.771 \pm 0.05$ | $0.779 \pm 0.06$ | $0.814 \pm 0.05$ | $0.769 \pm 0.05$ | $0.751 \pm 0.05$ | $0.746 \pm 0.06$ | $0.744 \pm 0.06$ |
| SMGRU-ODE(H=1) | $0.817 \pm 0.08$ | $0.791 \pm 0.08$ | $0.766 \pm 0.07$ | $0.766 \pm 0.08$ | $0.813 \pm 0.05$ | $0.777 \pm 0.05$ | $0.749 \pm 0.05$ | $0.743 \pm 0.05$ | $0.744 \pm 0.07$ |
| $\mathcal{L}_{cls}$ | $0.854 \pm 0.06$ | $0.827 \pm 0.06$ | $0.795 \pm 0.04$ | $0.799 \pm 0.04$ | $0.832 \pm 0.06$ | $0.784 \pm 0.05$ | $0.756 \pm 0.04$ | $0.764 \pm 0.05$ | $0.715 \pm 0.05$ |
| $\mathcal{L}_{cls} + \mathcal{L}_{rank}$ | $0.852 \pm 0.05$ | $0.828 \pm 0.04$ | $0.803 \pm 0.01$ | $0.812 \pm 0.02$ | $\mathbf{0.846 \pm 0.04}$ | $0.788 \pm 0.05$ | $0.773 \pm 0.02$ | $0.781 \pm 0.02$ | $0.769 \pm 0.04$ |
| $\mathcal{L}_{cls} + \mathcal{L}_{rank} + \mathcal{L}_{cns-rsk}$ | $\mathbf{0.857 \pm 0.05}$ | $0.832 \pm 0.04$ | $0.807 \pm 0.04$ | $0.810 \pm 0.03$ | $0.834 \pm 0.04$ | $0.793 \pm 0.03$ | $0.774 \pm 0.03$ | $0.776 \pm 0.03$ | $0.773 \pm 0.05$ |
| Proposed | $0.856 \pm 0.05$ | $\mathbf{0.844 \pm 0.04}$ | $\mathbf{0.819 \pm 0.02}$ | $\mathbf{0.822 \pm 0.03}$ | $0.840 \pm 0.05$ | $\mathbf{0.818 \pm 0.04}$ | $\mathbf{0.800 \pm 0.04}$ | $\mathbf{0.803 \pm 0.04}$ | $\mathbf{0.777 \pm 0.04}$ |

subject regularization, hence require training a single branch network. The results in Table 2 indicate the superiority of our proposed method which outperforms the existing methods at all time-points. SODEN, another N-ODE-based method showed signs of overfitting with good performance on the validation set (for selecting the best-performing models in each fold) but led to a drastic drop in performance on the test sets across all folds.

Table 2: Comparison with State-of-the-Art. Best performance is highlighted in bold.

| | AUROC | | | | Balanced Accuracy | | | | |
| | 6 | 12 | 18 | 24 | 6 | 12 | 18 | 24 | C-index |
|---|---|---|---|---|---|---|---|---|---|
| Cens. Cross-Entropy | $0.787 \pm 0.06$ | $0.779 \pm 0.06$ | $0.776 \pm 0.05$ | $0.789 \pm 0.04$ | $0.764 \pm 0.05$ | $0.739 \pm 0.04$ | $0.731 \pm 0.03$ | $0.741 \pm 0.02$ | $0.767 \pm 0.04$ |
| Logistic Hazard | $0.787 \pm 0.06$ | $0.787 \pm 0.04$ | $0.779 \pm 0.04$ | $0.797 \pm 0.03$ | $0.780 \pm 0.06$ | $0.766 \pm 0.03$ | $0.745 \pm 0.04$ | $0.755 \pm 0.04$ | $0.769 \pm 0.04$ |
| DeepSurv | $0.755 \pm 0.13$ | $0.735 \pm 0.12$ | $0.720 \pm 0.11$ | $0.728 \pm 0.12$ | $0.734 \pm 0.12$ | $0.702 \pm 0.10$ | $0.681 \pm 0.09$ | $0.679 \pm 0.09$ | $0.768 \pm 0.04$ |
| SODEN | $0.673 \pm 0.09$ | $0.707 \pm 0.05$ | $0.703 \pm 0.04$ | $0.721 \pm 0.05$ | $0.676 \pm 0.05$ | $0.691 \pm 0.03$ | $0.685 \pm 0.04$ | $0.698 \pm 0.04$ | $0.710 \pm 0.05$ |
| Proposed | $\mathbf{0.856 \pm 0.05}$ | $\mathbf{0.844 \pm 0.04}$ | $\mathbf{0.819 \pm 0.02}$ | $\mathbf{0.822 \pm 0.03}$ | $\mathbf{0.840 \pm 0.05}$ | $\mathbf{0.818 \pm 0.04}$ | $\mathbf{0.800 \pm 0.04}$ | $\mathbf{0.803 \pm 0.04}$ | $\mathbf{0.777 \pm 0.04}$ |

## 4. Conclusion

A wide variability in progression speed and the lack of well-established biomarkers make predicting the progression of AMD challenging. We proposed a novel framework that combines an AMD stage classifier with a N-ODE to forecast dAMD onset at continuous future times. To learn meaningful features from scarce labels, we enforce (i) intra-subject consistency to ensure that the feature embedding is sensitive to temporal changes in the retina to predict the future; (ii) temporal ordering, where a scan's proximity to the AMD classifier's decision hyperplane is inversely related to its time-to-conversion. These constraints enabled our model to outperform several existing deep survival analysis methods. Additionally, temporal ranking allowed us to derive a scalar risk score to stratify eyes into low and high risk groups. While training uses longitudinal OCT scans, only a single scan at test time is needed for future conversion prediction. Our method for predicting dAMD onset can facilitate patient-specific disease management and enrich clinical trial populations with high-risk patients. Currently, the proposed method has been evaluated on a single-center dataset. Further evaluation of our method on multi-center data and adaptation to other survival analysis tasks in the medical domain, such as progression-free survival in cancer patients, are potential directions for future work. Use of a ViT based encoder and incorporating segmentations of relevant retinal layers and lesions as additional inputs may also be considered in the future to further improve performance.

## Acknowledgments

This research was funded in whole, or in part, by the Austrian Science Fund (FWF) [10.55776/FG9], and Wellcome Trust Collaborative Award Ref. 210572/Z/18/Z.

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

## Appendix A. Intuitive Explanation of the Methodology

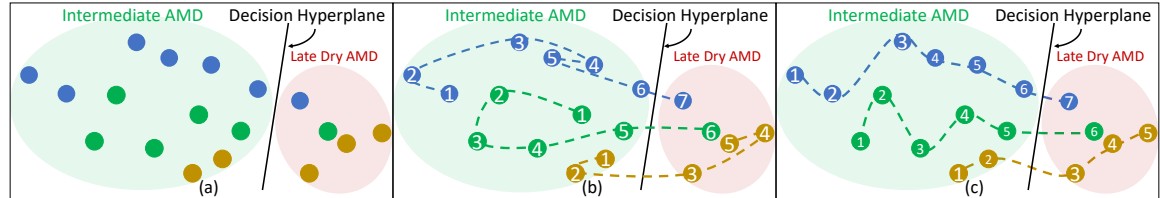

Figure 3: (a) The AMD stage classifier learns a decision hyper-plane separating iAMD from dAMD. Each scan is considered to be an independent sample. The learned feature should capture pathologies that distinguish iAMD from dAMD. (b) A N-ODE is introduced along with the stage classifier. The N-ODE traces the trajectory of disease progression (shown as dotted lines connecting the points of the same color, representing scans coming from the same eye at different time-points). Now the feature also needs to capture the subtle retinal changes indicative of the future disease state. (iii) A notion of direction is incorporated in the feature embedding. The closer a point is to the decision hyperplane, the smaller its time-to-conversion. AMD being an irreversible disease can only progress in time, so scans from a later visit of an eye (shown by numbered indices) have to successively get closer to the decision hyperplane.

## Appendix B. Implementation Details

All experiments were performed in Python 3.8.16 with Pytorch 2.0.0. The proposed method was trained with batches comprising 16 image pairs for 200 epochs (300 batch updates per epoch), using the MTAdam (Malkiel and Wolf, 2021) optimizer for dynamic loss tuning. A cyclic learning rate scheduler was employed with a minimum and maximum learning rate

of $10^{-6}$ and $10^{-4}$ respectively. The performance on the validation set was monitored at the end of each epoch for early stopping with a patience of 50 epochs. The N-ODE was implemented with the *torchdiffeq* library (Chen, 2018). The Euler method was used as the ODE-solver due to its computational efficiency with a step size of 0.06, where the time between 0-3 years was mapped to [0,1]. During training, each batch was constructed with random training image-pairs $(\boldsymbol{I}_j, \boldsymbol{I}_k)$ with a time-interval of 0-3 years between them. The training batches were constructed to ensure that all $\boldsymbol{I}_j$ were in the iAMD stage while half of the $\boldsymbol{I}_k$ in each batch were in the dAMD stage (through oversampling) to enable the training of the AMD stage classifier.

The proposed method required around 6 GB of GPU memory to train using a training batch size of 16 image pairs. The ConvNeXt-Tiny (Liu et al., 2022) encoder was initialized with the standard Image-Net pre-trained weights for end-to-end fine-tuning. The proposed SMGRU-ODE model with D=3, H=12 has 4,798,848 learnable network parameters.

## Appendix C. Eye-level Performance Comparison with Bootstrapping:

Table 3: Eye-level Bootstrap Performance (mean ± std. dev.). Best values in each column is highlighted in bold.

| | AUROC | | | | Balanced Accuracy | | | | |
| | 6 | 12 | 18 | 24 | 6 | 12 | 18 | 24 | C-index |
|---|---|---|---|---|---|---|---|---|---|
| Proposed | **0.863 ± 0.10** | **0.827 ± 0.10** | **0.808 ± 0.07** | **0.816 ± 0.07** | **0.871 ± 0.11** | **0.811 ± 0.09** | **0.789 ± 0.07** | **0.801 ± 0.06** | **0.769 ± 0.06** |
| Cens. Cross-Entropy | 0.775 ± 0.14 | 0.772 ± 0.103 | 0.773 ± 0.10 | 0.790 ± 0.08 | 0.804 ± 0.11 | 0.756 ± 0.11 | 0.742 ± 0.07 | 0.746 ± 0.06 | 0.762 ± 0.06 |
| Logistic Hazard | 0.769 ± 0.19 | 0.768 ± 0.12 | 0.763 ± 0.09 | 0.786 ± 0.08 | 0.792 ± 0.14 | 0.760 ± 0.11 | 0.749 ± 0.08 | 0.766 ± 0.08 | 0.749 ± 0.08 |
| DeepSurv | 0.769 ± 0.18 | 0.710 ± 0.16 | 0.712 ± 0.14 | 0.723 ± 0.14 | 0.749 ± 0.17 | 0.689 ± 0.12 | 0.682 ± 0.12 | 0.686 ± 0.12 | 0.752 ± 0.07 |
| SODEN | 0.675 ± 0.24 | 0.674 ± 0.17 | 0.673 ± 0.13 | 0.698 ± 0.11 | 0.711 ± 0.19 | 0.671 ± 0.14 | 0.665 ± 0.11 | 0.693 ± 0.10 | 0.673 ± 0.09 |

Eye-level bootstrapping involves multiple re-samplings of the test set in each fold. In each re-sampling, one OCT scan is selected from each eye (by randomly selecting any one of the patient visits). This re-sampling process is repeated 1000 times for each of the five folds to report the average performance across the $5 \times 1000 = 5000$ sample estimates across all folds (see Table 3).

## Appendix D. Preprocessing and Data Augmentation

The top and bottom boundaries delineating the retinal tissue called the Inner Limiting Membrane (ILM) and the Bruch's Membrane (BM) were extracted using the automated method in (Fazekas et al., 2022). Thereafter, the curvature of the retinal surface was flattened by shifting each A-scan by an offset such that the BM lies on a straight plane similar to (Emre et al., 2022). The five central B-scans centered around the fovea spanning 5 mm across the A-scans (image columns) were extracted and the region containing the retinal tissue between the ILM and BM was cropped with a margin of 280 micron in the bottom to include the choroid region and resized to $248 \times 248$. The intensity was linearly scaled to [-1,1].

During training, 3 consecutive B-scans (slices) out of the 5 central B-scans extracted during preprocessing were randomly selected from each scan and provided as input to the

ConvNeXt-Tiny model in place of the three RGB color channels. The data augmentations during training involved random translations, horizontal flip, random crop-resize, Gaussian noise, random in-painting and random intensity transformations.

During inference, no data augmentation was employed. Of the 5 central B-scans extracted, 3 sets of images were constructed, each using 3 consecutive B-scans as channels similar to RGB in natural images (and the average predictions from these 3 images was used). The same approach was also employed for evaluating the other state-of-the-art methods for comparison.

## Appendix E. Dynamic Loss Tuning

Determining the value of the tunable loss weights $\lambda_1$, $\lambda_2$, $\lambda_3$ and $\lambda_4$ in Eq. 3 through a systematic grid search is computationally expensive as it requires training multiple model configurations. Instead, we used Multi-Term Adam (MTAdam) (Malkiel and Wolf, 2021) to dynamically adapt the loss weights during training. MTAdam extends the ADAM optimizer by tracking derivatives and the first and second order moments of each loss term separately and continuously balances their gradient magnitudes across all layers during training batch updates. To evaluate the impact of this design choice, we retrained the model with different alternatives presented below in Table 4. In case of *Equal Weighting* we fixed all weights to $\lambda_1 = \lambda_2 = \lambda_3 = \lambda_4 = 1.0$. In case of *Handcrafted weights*, we fixed $\lambda_1 = 1.0$, $\lambda_2 = 0.1$, $\lambda_3 = 1.0$ and $\lambda_4 = 10.0$ by observing the scale and the perceived relative importance of the different loss terms. The uncertainty weighting based method in (Kendall et al., 2018) is another alternative automatic method for dynamic loss tuning which was used along with the modifications proposed in (Liebel and Körner, 2018) to avoid the loss becoming negative during training.

Table 4: Comparison of different loss weighting strategies (mean $\pm$ std. dev.). Best values in each column is highlighted in bold.

| | AUROC | | | | Balanced Accuracy | | | | |
| --- | --- | --- | --- | --- | --- | --- | --- | --- | --- |
| | 6 | 12 | 18 | 24 | 6 | 12 | 18 | 24 | C-index |
| Equal weighting | $\mathbf{0.862 \pm 0.07}$ | $0.828 \pm 0.04$ | $0.802 \pm 0.03$ | $0.807 \pm 0.04$ | $\mathbf{0.855 \pm 0.05}$ | $0.798 \pm 0.03$ | $0.777 \pm 0.02$ | $0.775 \pm 0.02$ | $0.772 \pm 0.03$ |
| Handcrafted weights | $0.843 \pm 0.06$ | $0.835 \pm 0.04$ | $0.808 \pm 0.03$ | $0.820 \pm 0.02$ | $0.831 \pm 0.05$ | $0.796 \pm 0.04$ | $0.780 \pm 0.04$ | $0.793 \pm 0.04$ | $0.772 \pm 0.02$ |
| MT-ADAM | $0.856 \pm 0.05$ | $\mathbf{0.844 \pm 0.04}$ | $\mathbf{0.819 \pm 0.02}$ | $\mathbf{0.822 \pm 0.03}$ | $0.840 \pm 0.05$ | $\mathbf{0.818 \pm 0.04}$ | $\mathbf{0.800 \pm 0.04}$ | $\mathbf{0.803 \pm 0.04}$ | $\mathbf{0.777 \pm 0.04}$ |
| Uncertainty weighting | $0.843 \pm 0.06$ | $0.819 \pm 0.04$ | $0.790 \pm 0.03$ | $0.797 \pm 0.03$ | $0.818 \pm 0.07$ | $0.773 \pm 0.04$ | $0.749 \pm 0.04$ | $0.752 \pm 0.04$ | $0.763 \pm 0.04$ |

## Appendix F. Identification of Risk Groups

We calibrated the risk scores in each fold to lie in the $[0, 1]$. This was performed with bicubic interpolation to map the $x^{th}$ percentile of the risk scores in the validation set to $\frac{x}{100}$ (e;g., the $10^{th}$ percentile of the risk scores is mapped 0.1 and so on). The test set predictions of the calibrated risk scores were combined from the five folds to obtain a risk score for each OCT scan. The scans were then stratified into 3 groups with low risk ($0 \leq r \leq 0.33$), moderate risk ($0.33 < r \leq 0.67$) and high risk ($0.67 < r \leq 1$). A population-level survival function for these groups is plotted in Fig. 4 (a) using the Kaplan–Meier estimator on the GT conversion time. It depicts the mean and standard deviation of the survival probability

for each population group, computed across 1000 re-samplings using bootstrapping. The survival curves for the three risk groups show a clear separation, thereby demonstrating the effectiveness of the proposed risk score. The learned feature embedding (Fig. 4 (b)) exhibit a smooth transition from fast(red) to slow converters(blue) along the feature manifold where the gray dots represent the censored scans.

The Saliency maps obtained for the risk scores in Fig. 5 show the network to be sensitive to the structural changes around the RPE (e.g. Fig.5 (a),(b)) and Hyperreflective Foci (HRF) (e.g. Fig.5 (d),(e) ) which have been clinically linked to dAMD progression.

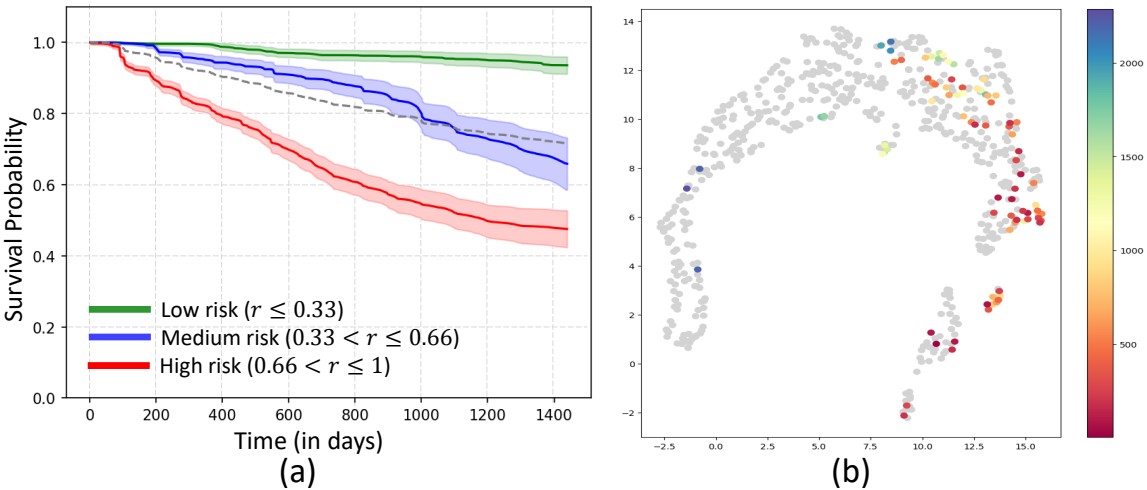

Figure 4: (a) Kaplan-Meier curves for different risk groups; (b) UMAP plot of feature embedding for one of the five folds. The censored scans are depicted with gray dots and the converters colored by their time to conversion (red indicates fast conversion)

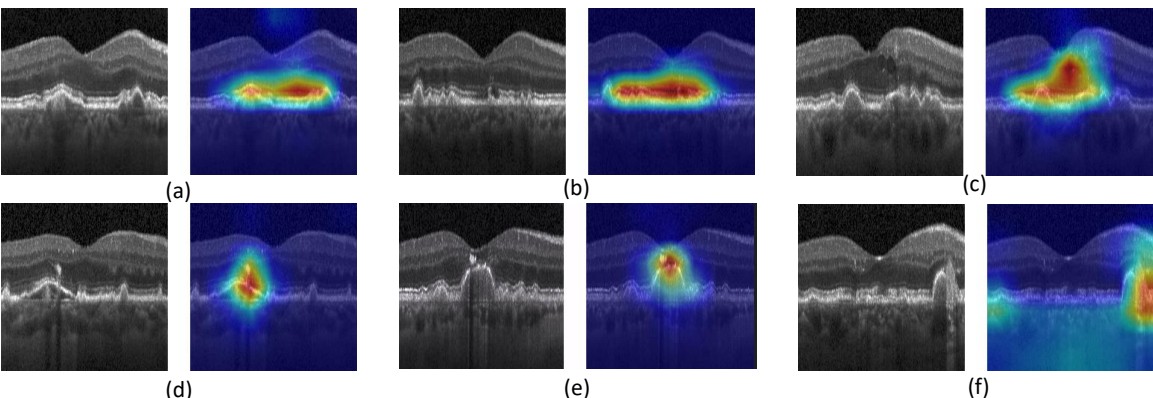

Figure 5: Grad-CAM Saliency maps for the risk score.

