# OpenReview forum: "Predicting Age-related Macular Degeneration Progression from Retinal Optical Coherence Tomography with Intra-Subject Temporal Consistency"
_MIDL.io/2024/Conference — MIDL 2024 Oral_

### Official Review · Reviewer_z4TP · 2024-02-28

**Confidence:** 4
**Preliminary Rating:** 5
**Recommendation:** Oral
**Final Rating:** 5

**Summary:**

The authors train a classifier to discriminate between intermediate and dry stages of Age-related macular degeneration (AMD), modeling the future trajectory of the disease progression in the learned embedding space.
Interesting and well motivated constraints are imposed, including a temporal ordering so that the distance of a scan from the decision hyperplane of the AMD stage classifier is inversely related to its time-to-conversion. An intra-subject temporal consistency is also included.
The proposed method is evaluated on a middle-size dataset (235 eyes) with AUC of predicting conversion within the next 2 years, and C-index. Extensive results, comparisons and analyses are provided.

**Strengths:**

I found the paper very interesting. A solid framework is proposed, with a well motivated application and different well suited components/losses (modeling of CDF, siamese architecture, consistency losses, risk score ranking), well balanced with dynamic loss weighting. The benefit of the different modules and losses is shown with ablation studies.
The proposed approach largely outperforms state of the art methods on a middle-size dataset.
Extensive results and analyses are provided including Kaplan-Meier curves, UMAP and (a single) grad-CAM saliency.

**Weaknesses:**

Nothing major. Figure 1 could be more self-explanatory, mentioning the main blocks and losses for a quick overview of the method (without making it too complicating).
No statistical tests are performed.

**Detailed Comments:**

Make sure to define all abbreviations (e.g. AMD in abstract, CDF)

Why is concordance index abbreviated CCI?

A few minor typos, e.g. "predicts the logits r j which provides", "Since, the ...", "where scalar w and b" to fix.

It would be great to evaluate the proposed approach on other survival tasks in future work, e.g. progression free survival in cancer patients etc.

In the dataset description, you mention 235 eyes. Do some originate from the same patients ? If so, do you ensure they are in the same fold ?

**Justification Of Final Rating:**

I still think that minor improvements can be made, and that a response to all reviewers is important. However, my original recommendation is unchanged. I hope the authors will update the manuscript. Reviewing requires time (here 3 reviewers + AC), and it is important to incorporate this feedback even if the paper is likely accepted.

**Justification Of The Preliminary Rating:**

The paper is clear and well written. The method is well motivated, with an interesting design and well suited for the task.

The various results largely support the hypotheses and the proposed method.

**Questions To Address In The Rebuttal:**

Nothing major, fix the minor comments and make the figure more self-explanatory.

**Special Issue:**

Yes

---

> ### Author Response · Authors · 2024-03-17
>
> We thank the reviewer for recognizing the value of our contribution.  In the revised manuscript, we have made the following changes to address the reviewer’s comments.
>
> 1. We have now modified Fig. 1 as suggested by the reviewer to include the losses and extended the caption to make it more self-explanatory.
>
>
> 2. We agree with the reviewer that the proposed method should be evaluated on other survival tasks in the medical domain in the future.  We have now mentioned it at the end of the Conclusion section.
>
>
> 3. We have ensured that the abbreviations are defined at the place of their first use such as Age-related Macular Degeneration (AMD) in the Abstract  and Cumulative Distribution Function (CDF) in the introduction section.
>
>
> 4. We have now also fixed the minor typos pointed out by the reviewer.
>
> 5. We have modified the abbreviation used for Concordance Index from CCI in the earlier version of the manuscript to C-index.

---

### Official Review · Reviewer_EbrR · 2024-02-28

**Confidence:** 5
**Preliminary Rating:** 5
**Recommendation:** Oral
**Final Rating:** 5

**Summary:**

The goal of this work is to develop an approach to predict the risk of age-related macular degeneration (AMD) from optical coherence tomography (OCT). They propose to combine an AMD stage classifier and a novel architecture using stacked multi-head GRU and a neural ODE to exploit intra-patient longitudinal information. OCT images. A siamese architecture trained on random image pairs from the same patient learn embeddings, a disease classifier and a prediction of the trajectory thanks to the N-ODE. The database consists in more than 3500 scans from one center. The validation of the approach through the ablation study and results (outperforming state of the art) are really convincing.

**Strengths:**

The paper is strong:
- clinical relevance is clear and the proposed solution yields excellent results.
- careful and in depth-analysis. Process is sufficiently documented to be reproducible.
- considering the time-to-conversion as continuous is original and the corresponding architecture smart and interesting. This could be applied in other contexts.

**Weaknesses:**

The work does not have any significant weaknesses:
- redaction could be improved (see detailed comments) to correct some small misprints and make it easier to read.
- this is a single center study that would have to be confirmed in a multi-center study.

**Detailed Comments:**

On redaction:
- Key acronyms should be defined at their first use, esp in title and abstract (AMD, OCT) for reader's convenience.
- There are also small errors e.g.
   - on page 5 "as time progresses  from t to t + dt
by an infinitesimal amount, f (t) is displaced by the instantaneous velocity vector $v_D$". This should be $v_D dt$.
   - On page 6  "During forward pass of the N-ODE, the feature for a future time-point k is given by...", $v_D$ does not directly depend on $t$, it should read $v_D(f_{t+j})$.
- Fig 2 is difficult to read, it could be made clearer (perhaps by limiting the notations).

Minor questions for reproducibility:
- Was the accuracy threshold optimized or is it set to 0.5?

**Justification Of Final Rating:**

I would have appreciated a short revision to address my concerns on the redaction at least. Yet I stand by my initial rating on this work that deserves to be considered for acceptance : strong accept.

**Justification Of The Preliminary Rating:**

This is solid work with an interesting approach for a relevant clinical problem. It presents on novel architecture that could be adapted in other contexts where one has access to longitudinal evolution of a disease and wishes to predict a similar degradation. Overall the paper is well-written and the reproducibility satisfactory.

**Questions To Address In The Rebuttal:**

This would not change my mind, but some perspectives could be given in the conclusion.

**Special Issue:**

Yes

---

> ### Author Response · Authors · 2024-03-17
>
> We are grateful to the reviewer for recognizing the merit of our paper. In the revised manuscript, we have made the following changes based on the reviewer's valuable suggestions.
>
>
> 1. We have expanded our conclusion with a discussion of some limitations of the current work which provide us with possible directions for extending the current work. This includes adapting our method on other survival analysis tasks in the medical domain and evaluating our current method on datasets acquired across multiple sites as pointed out by the reviewer.
>
>
> 2. The Balanced Accuracy was computed by selecting an optimal threshold on the validation set in each fold.  This has been clarified in the revised manuscript.
>
>
> 3. We have defined the full-form of the acronyms AMD and OCT both in the paper’s title  and Abstract as suggested by the reviewer to improve clarity.
>
>
> 4. We have made minor modifications in the notations used in Fig 2 as suggested by the reviewer to improve readability.
>
>
> 5. We are grateful to the reviewer for pointing out the mistakes in the mathematical notations on pg. 5 and pg. 6 which we have now corrected in the revised manuscript.

---

### Official Review · Reviewer_D1DK · 2024-02-29

**Confidence:** 4
**Preliminary Rating:** 4
**Recommendation:** Oral
**Final Rating:** 3.5

**Summary:**

The paper explores disease progression using neural ODEs and makes a number of technical contribution paired with good experimental validation to predict iAMD to dAMD progression.

**Strengths:**

Very promising results compared to prior work. Comprehensive evaluation using both ablation studies for algorithmic choices (loss terms, automatic hyperparamters etc.) and against SOTA work. Overall good description despite the topic being quite complex.

**Weaknesses:**

The relevance of censoring is mentioned in the beginning but not clearly resolved during the method and/or experimental section.
The result presentation is slightly underwhelming, the numerical tables have poor readability (very small font) and distribution plots or similar are missing (apart from Appendix Fig. 4). Some more qualitative results could be added there as well.

**Detailed Comments:**

I wonder whether more reasoning for the choice of convolutional backbone could be given: this seems to be a very compact model whereas the SMGRU-ODE is quite parameter heavy.
I was slightly surprised not to see any mentioning of transformer models to predict the temporal progression (e.g. http://proceedings.mlr.press/v146/hu21a/hu21a.pdf) . Are neural-ODEs more adapt to smaller datasets?
The contribution / difference to prior work is a bit hidden: to my understanding neither one element (N-ODE, image-based prediction, AMD progression, ranking loss) is on its own new but the combination of these pieces make the proposed method strong.

**Justification Of Final Rating:**

Since, they were a number of small concerns that I raised during the review, I am not very happy about not receiving a response. Nevertheless their seems to be an agreement that the paper should be accepted. So I hope the authors will incorporate some improvements in a final version

**Justification Of The Preliminary Rating:**

This is a very solid paper which nicely fits within the scope of the MIDL conference. It has only minor points that may require improvement. I am confident the authors can address my questions during the rebuttal.

**Questions To Address In The Rebuttal:**

Some sentences/statements remain unclear: e.g. parametric models like CoxPH face limitations in hazard function distribution modelling flexibility. This is really hard to grasp for a non-expert Isn't the combination of parametric models with deep networks more capable of flexibel modelling? Furthermore, the concept of hybrid models (ones that extract segmentations from images) seems very useful but is not explored: could the authors envision further gains by incorporating this additional expert supervision?

---

> ### Author Response · Authors · 2024-03-17
>
> We thank the reviewer for helpful suggestions and constructive feedback. Detailed discussion follows.
>
> 1. Vision Transformers (ViT) are known to require large datasets and suitable pre-training to avoid over-fitting. Moreover many ViT architectures are resource intensive with their computational complexity and memory requirements increasing quadratically with the number of input tokens making them unsuitable for our preliminary experiments with Neural-ODEs. However, we agree with the reviewer that Ablation experiments with different encoder architectures including ViTs need to be carried out as a part of future work which we have now mentioned at the end of the Conclusion Section.
>
>
> 2. Reasoning behind using the ConvNeXt-Tiny architecture as backbone: ConvNext-Tiny with 29M network parameters and 4.5GFLOPs is comparable to ResNet-50 (25.6M parameters and 4.1GFLOPs). Hence, it is a reasonable choice for our medium sized dataset. Moreover, ConvNext-Tiny has been shown to outperform some of the ViT models of similar size on the Image-Net classification task such as Swin-Tiny, DeiT-Small and DeiT-Big. Compared to the ConvNext-Tiny Encoder, our proposed Multi-head GRU-ODE architecture is relatively small with 4.79M parameters. (in contrast to 29M for ConvNeXt-Tiny). We have now mentioned the motivation to use ConvNeXt-Tiny at the beginning of page 4 under the “AMD stage Classifier” heading.
>
>
> 3. We thank the reviewer for pointing out a relevant work on survival analysis based on the Transformer architecture by Hu et. al. which we have now added to our related work.
>
>
> 4.  “parametric models like CoxPH face limitations in hazard function distribution modelling flexibility” In DeepSurv and similar deep learning extensions of the CoxPH model, a single baseline cumulative hazard function $H_0(t)$ is computed for the entire training dataset using the Breslow Estimator. The cumulative hazard function for a specific eye with scan I, denoted as $H(t|I)$, is then determined as $H(t|I) = H_0(t) * \exp(r)$, where $r$ is a constant scaling factor across time. This means that $H_{0}(t)$ serves as a "prior" cumulative hazard across the population, scaled by $\exp(r)$ to generate individual cumulative hazard functions for each eye. Despite predicting $r$ with Deep Learning, the underlying assumption remains that hazard functions are scaled versions of $H_0(t)$. This constraint limits the flexibility of deep CoxPH models, as real-world hazard functions for different individuals may follow various monotonically increasing shapes, not necessarily scaled versions of a common $H_0(t)$.  In the revised manuscript we have now modified the line “parametric models like CoxPH face limitations in hazard function distribution modelling flexibility” to “Although CoxPH has been extended with DL using images \cite{katzman2018deepsurv}, they have not yet been explored to model AMD progression so far. Moreover, these models are inflexible as each patient's hazard function is constrained to be a scaled version of the same baseline hazard across the entire population.”
>
>
> 5. “Furthermore, the concept of hybrid models (ones that extract segmentations from images) seems very useful but is not explored: could the authors envision further gains by incorporating this additional expert supervision?”     We agree with the reviewer's suggestion regarding potential performance improvement through the inclusion of additional inputs such as retinal layer segmentations and lesions, which we have now mentioned as part of future work in the conclusion section. However, this extension poses challenges, as it depends on the development of accurate segmentation networks for retinal layers and lesions in AMD patients. These networks may encounter segmentation errors and require collecting pixel-level segmentation Ground Truth for training. Additionally, the specific retinal layers and lesions that indicate future conversion risk remain clinically unclear, raising questions about which biomarkers should be prioritized. Furthermore, meaningful architectural adjustments are necessary in the encoder backbone to effectively process multiple inputs, including raw OCT scans and segmentation maps.
>
>
> 6. Additional qualitative results in the form of Gradcam Saliency Maps have been appended to the Appendix , Fig. 5.

---

### Comment · Area_Chair_McBv · 2024-03-15
**Gentle reminder of rebuttal deadline**

Dear authors,

This is just a gentle reminder that the rebuttal deadline is coming up. I encourage you to take this opportunity so you can engage with the reviewers in the upcoming discussion phase.

Best wishes,
AC

---

### Meta-Review · Area_Chair_McBv · 2024-04-02

**Recommendation:** Accept (Oral)
**Confidence:** 5

**Metareview:**

This paper proposes a framework to predict AMD progression using neural ODEs. The paper was very well received and was recommended for oral presentation by all reviewers. Regrettably, the authors did not provide a rebuttal. Nevertheless, the reviewers mostly kept their original score and I continue to accept the paper for presentation at MIDL.

I strongly urge the authors to incorporate the reviewer comments in the camera-ready version and respond to reviewer comments in the future. I thank the reviewers for their valuable help.

---

### Decision · Program_Chairs · 2024-04-06

Accept (Oral)

---

> ### Author Response · Authors · 2024-04-07
>
> Dear Program Chairs,
>
> During the rebuttal phase, I had selected the wrong settings by mistake and my replies were not visible to the reviewers. I realized it now after seeing the final response from the reviewers. I apologize for this mistake and I have now edited the settings so that everyone could see the replies. Please suggest  what additional steps could be taken now to address this mistake.
>
> Thanks and Best Regards,
> Arunava Chakravarty

---

> > ### Author Response · Authors · 2024-04-07
> > **comments to reviewers**
> >
> > Dear authors,
> >
> > The PCs had access to all your answers, and they considered this for the final decision of the paper. Now, your answers are public to everyone, and no additional steps are required.
> >
> > Thank you for your submission,
> > Maria, on behalf of the organization team